# Phenotypic Analysis of Hematopoietic Stem and Progenitor Cell Populations in Acute Myeloid Leukemia Based on Spectral Flow Cytometry, a 20-Color Panel, and Unsupervised Learning Algorithms

**DOI:** 10.3390/ijms25052847

**Published:** 2024-02-29

**Authors:** Thomas Matthes

**Affiliations:** 1Hematology Service, Oncology Department, University Hospital Geneva, Rue Gabrielle Perret-Gentil, 1205 Geneva, Switzerland; thomas.matthes@hug.ch; 2Clinical Pathology Service, Diagnostics Department, University Hospital Geneva, Rue Gabrielle Perret-Gentil, 1205 Geneva, Switzerland

**Keywords:** hematopoiesis, stem cells, leukemia, acute myeloid leukemia, myelodysplastic syndrome, measurable residual disease, unsupervised analysis, flow cytometry

## Abstract

The analysis of hematopoietic stem and progenitor cell populations (HSPCs) is fundamental in the understanding of normal hematopoiesis as well as in the management of malignant diseases, such as leukemias, and in their diagnosis and follow-up, particularly the measurement of treatment efficiency with the detection of measurable residual disease (MRD). In this study, I designed a 20-color flow cytometry panel tailored for the comprehensive analysis of HSPCs using a spectral cytometer. My investigation encompassed the examination of forty-six samples derived from both normal human bone marrows (BMs) and patients with acute myeloid leukemia (AML) and myelodysplastic syndromes (MDS) along with those subjected to chemotherapy and BM transplantation. By comparing my findings to those obtained through conventional flow cytometric analyses utilizing multiple tubes, I demonstrate that my innovative 20-color approach enables a more in-depth exploration of HSPC subpopulations and the detection of MRD with at least comparable sensitivity. Furthermore, leveraging advanced analytical tools such as t-SNE and FlowSOM learning algorithms, I conduct extensive cross-sample comparisons with two-dimensional gating approaches. My results underscore the efficacy of these two methods as powerful unsupervised alternatives for manual HSPC subpopulation analysis. I expect that in the future, complex multi-dimensional flow cytometric data analyses, such as those employed in this study, will be increasingly used in hematologic diagnostics.

## 1. Introduction

Hematopoiesis represents a milestone as the first system where tissue-specific stem cells, known as the hematopoietic stem cells (HSCs), were identified. Initially, it was thought that cells expressing the surface marker CD34 corresponded to these HSCs; however, recent advancements have revealed that the CD34pos cell population is remarkably heterogeneous, comprising various subpopulations of progenitor and precursor cells collectively termed Hematopoietic Stem and Progenitor Cells (HSPCs). A mere fraction—less than 1%—of these CD34pos cells has been substantiated as authentic HSCs through in vivo mouse transplantation assays [1,2,3]. Other distinct subpopulations exhibit varying maturation stages and differentiation potential and show the beginnings of surface marker expressions of diverse cell lineages such as B-lymphoid, erythroid/megakaryocytic, and myelomonocytic.

In various pathological conditions such as acute leukemias and myelodysplastic syndromes (MDS), the integrity of the CD34pos HSPC compartment is disrupted. In acute leukemias, the presence of malignant blasts contributes to the suppression of all normal HSPC subpopulations. Conversely, in MDS, there is a reduction in HSPC heterogeneity, characterized by diminished CD19pos hematogones and alterations in the relative composition of various myeloid progenitors and precursors [4,5,6,7]. Post bone marrow transplantation, HSPC reconstitution frequently exhibits heterogeneity and remains incomplete. This is attributed to deficiencies in supportive stromal cells and the immunosuppressive or cytotoxic effects of the multiple drugs administered to patients during and after the transplantation procedure.

Different strategies have been developed in flow cytometry to distinguish HSPC subpopulations. Notably, several studies have identified the CD38neg CD45RAneg subset of HSPCs as encompassing this genuine HSC population [8]. Other markers employed for a more precise definition of this population include CD90 and CD49f [9,10]. Further characterization involves the identification of more mature CD34pos HSPC subpopulations, such as multipotent progenitors (MPPs), common myeloid progenitors (CMPs), megakaryocyte-erythroid progenitors (MEPs), granulocyte-monocyte progenitors (GMPs), and common lymphoid progenitors (CLPs). These subpopulations are discerned based on the presence of various combinations of CD38, CD33, CD45RA, CD123, and CD19 marker expressions [11,12,13,14,15].

Similar to the analysis of normal HSPCs, flow cytometry is used for the detection and characterization of malignant leukemic blasts. In fact, the comparison of surface marker expressions between malignant blasts and normal HSPCs is pivotal as the identification of one or several aberrantly expressed antigens can substantiate the malignancy of a CD34pos cell population. The presence of 0.1% CD34pos cells exhibiting aberrant marker expression serves as a criterion for Measurable Residual Disease (MRD) positivity, as endorsed by the European LeukemiaNet (ELN) [16].

In efforts to enhance the sensitivity of malignant blast detection, various research groups have concentrated on the analysis of CD34pos CD38neg leukemic stem cells (LSCs). Like normal HSCs, these cells are supposed to constitute the real stem cells of a malignant blast population [17]. CD123 has been identified as being consistently expressed in LSCs from 16 out of 18 AML patients, being notably absent in normal HSCs [18]; CD371 expression was detected in LSCs from 68 out of 74 AML samples [19]. Another approach involved the design of a 13-color panel incorporating a cocktail of six markers in a single fluorescence for the analysis of aberrant markers in LSCs. This strategy demonstrated comparable sensitivity to the standard multiple-panel analysis, as reported in the study by Zeijlemaker et al. [20]. Still other strategies include those based on CD45RA [21] or concomitant CD371 and CD45RA detection on LSCs [22].

In recent years, significant strides in flow cytometry technology have paved the way for the concurrent evaluation of a greater number of parameters in single analyses. This progress has empowered researchers to characterize low-frequency cell populations and subpopulations in a highly correlated manner. Two groundbreaking platforms, namely full spectrum flow cytometry (FSFC) introduced by Sony and Cytek Biosciences and cytometry by time of flight (CyTOF) by Fluidigm, have revolutionized the field by facilitating high-dimensional data acquisition from panels featuring 20 or more antibodies.

While both technology platforms possess distinct advantages and disadvantages in profiling low-frequency populations, studies have reported superior cell recovery and faster acquisition times with the FSFC approach [23,24]. Analyzing the data generated from these high-dimensional analyses involves the use of various software algorithms. These algorithms facilitate unsupervised clustering of the data, exemplified by techniques such as Flow Self Organizing Maps (FlowSOM), Spade, or Phenograph [25]. Additionally, they enable complexity reduction through dimensionality reduction methods like t-SNE (t-Distributed Stochastic Neighbor Embedding) or UMAP (Uniform Manifold Approximation and Projection), along with the creation of comprehensive graphical presentations [26,27].

It is anticipated that the exploration of HSPC subpopulations, employing a greater number of parameters beyond the capabilities of conventional 8- to 13-color flow cytometers, will facilitate a heightened level of precision and sensitivity when detecting MRD among patients with AML. Additionally, the incorporation of state-of-the-art software tools tailored for the analysis of these highly complex data is expected to further enhance the quality and accuracy of results.

In this study, I designed a 20-color panel tailored for the comprehensive analysis of HSPCs using a spectral cytometer. This investigation encompassed the examination of 46 samples derived from both normal human bone marrows and patients with AML and MDS along with those subjected to chemotherapy and BM transplantation. By comparing these findings to those obtained through conventional flow cytometric analyses utilizing multiple tubes, I demonstrate that this innovative 20-color approach enables a more in-depth exploration of HSPC subpopulations and the detection of MRD with at least comparable sensitivity. Furthermore, leveraging advanced analytical tools such as t-SNE and flowSom algorithms, extensive cross-sample comparisons are conducted. My results underscore the efficacy of these methods as powerful unsupervised alternatives for HSPC subpopulation analysis and MRD detection, thereby enhancing the depth and precision of this study.

## 2. Results

Detailed analysis of CD34pos HSPCs is important for MRD detection in the follow-up of acute leukemias, recognition of dysplasia in MDS, and BM reconstitution after chemotherapy and BM transplantation. A new 20-color panel was developed, analyzed by spectral cytometry, and compared to routine diagnosis based on several 10-color panels. Moreover, conventional manual 2D analysis was compared with automatic unsupervised clustering analysis methods.

### 2.1. Phenotypic Analysis of HSCPs Using Conventional and Spectral Cytometry

In routine analysis at the Flow Cytometry Diagnostics laboratory of the Geneva University Hospital, a Navios cytometer is used to determine the frequency of CD34pos HSPCs and to determine lymphoid CD19pos and CD33neg, and myeloid CD33pos/neg and CD19neg, HSPC subpopulations before examining their phenotype in detail (Appendix A). I performed similar analyses based on the newly developed 20-color panel and a Cytek spectral cytometer. Comparison of the quantitative values obtained by the two different approaches showed excellent correlation coefficients (CC > 0.98) for the percentage of CD34pos cells found in bone marrow samples from 46 patients (Appendix A) as well as for the myeloid/lymphoid HSPC ratio. Comparison of the frequencies of mature CD19pos B cells, CD3pos T cells, CD14pos monocytes, and CD123pos dendritic cell populations resulted in similarly excellent correlation coefficients (CC = 0.98, CC = 0.96, CC = 0.98, and CC = 0.97, respectively).

It was, therefore, concluded that the 20-color panel and the Cytek spectral cytometer could be used for the in-depth analysis of different HSPC subpopulations.

### 2.2. Phenotypic Analysis of Normal HSPC Subpopulations Based on Traditional Gating Strategies and the 20-Color Panel

Using the 20-color panel and a viability dye, an improved gating strategy could be defined for CD34pos HSPCs, which allowed me to distinguish myeloid from lymphoid HSPCs and evaluate the maturation of the myeloid HSPCs from the CD38neg state—containing the most immature HSCs—to the CD38pos state (Figure 1A).

Gating according to the CD38 marker expression does not allow the most immature myeloid HSPCs to be precisely defined since CD38 expression continuously increases from negative to strongly positive cells (Figure 2B; upper left dotplot). An additional strategy of HSPC subpopulation analysis was, therefore, used based on CD371 and CD45RA marker expressions (Figure 2B; upper right dotplot). In fact, CD371 (C-type lectin-like molecule-1; CLL-1) and CD45RA, an isoform of the CD45 complex, are differentially expressed in HSPC subpopulations and absent on the most immature HSC/MPP populations [19,28]. As depicted, using these two markers, HSPCs can be separated into four different subpopulations with the lower left quadrant (LLQ) containing the immature HSCs/MPPs (Figure 2B; upper left dotplot; brown color). The cell population in the LLQ was explored in more detail and it was found that the expression of the markers CD36 and CD200 allowed CD36neg and CD200pos HSCs/MPPs to be further distinguished from CD36pos and CD200neg MEPs. The analysis of the five normal BM samples showed very similar patterns in all of them and allowed the following mean frequency values to be established for the cell populations in the four quadrants as well as for the HSC/MPP population: LLQ: 41 ± 9%; ULQ: 16 ± 9%; URQ: 30 ± 4%; LRQ 13 ± 4%; and HSP/MPP: 14 ± 6% (Figure 2).

The phenotype of the different myeloid HSPC subpopulations visible in the CD371/CD45RA dotplot was then characterized in-depth based on all the different markers from the 20-color panel. This analysis made it possible to define cells in the LLQ as HSCs/MPPs and MEPs (CD36pos), cells between the LLQ and LRQ as CMPs, and cells in the LRQ and URQ as GMPs, with the beginnings of differentiation into GP and MP cells (CD38pos and CD64pos). The markers in the panel did not allow the unequivocal annotation of cells in the ULQ, which partially expressed CD200 and CD123. Interestingly, the cells with CD123 expression corresponding to pDC and basophil precursors were found in a distinct position between the ULQ and URQ (Appendix A).

Combining thus the CD34, CD38, CD45RA, and CD371 expression patterns clearly separated the HSPC subpopulations so that they could be easily analyzed (Figure 2B,C). Further differentiation into more mature hematopoietic cell populations can then be followed based on CD34neg precursors in the same data file, but this analysis lies outside the scope of this article.

The detailed analysis of myeloid HSPC maturation based on CD38 negative to strong expression showed that in all five normal BMs tested, HSCs/MPPs did not express the following 13 antigens: CD3, CD7, CD14, CD15, CD16, CD19, CD33, CD36, CD45RA, CD56, CD64, CD123, and CD371 (Figure 3A). The only antigens of the 20-color panel found to be expressed by HSCs/MPPs were CD34, CD117, CD133, CD200, and HLA-DR (Figure 3B). Thus, the beginnings of the differentiation of CD38pos HSPCs into myelo-monocytic precursors (CD64posCD371pos), erythroid precursors (CD36pos), plasmacytoid dendritic cells, and basophil precursors (CD123pos) could also be evaluated (Figure 3A).

It was, therefore, concluded that the 20-color panel allowed the detailed analysis of the frequencies and phenotypes of at least five different HSPC subpopulations present in normal bone marrow samples (CLPs, HSCs/MPPs, CMPs, MEPs, and GMPs) as well as the establishment of reference frequency values and extended reference phenotypes.

### 2.3. Analysis of Myeloid HSPC Subpopulations in Cases of MDS and AML and in Follow-Up Samples after Treatment

Based on the parallel analysis of CD38/all other panel markers and CD371/CD45RA dotplots, the frequencies and phenotypes of the different myeloid HSPC populations could then be analyzed in pathologic conditions. Nineteen samples were analyzed to address the question of MRD after chemotherapy and/or BM transplantation for AML. Routine flow analysis and molecular biology results showed concordantly absent MRD in 15/16 samples; Case 23 was considered negative based on flow cytometry but was found positive based on the molecular biology analysis, and three samples showed clearly positive MRD in flow, with 1.1% and 6.0% and 30% leukemic blasts, respectively (Cases 24–26).

With the 20-color panel tube, the fifteen samples with neg MRD based on routine cytometry, patterns similar to normal BMs were found in the CD371/CD45RA dotplots, with populations present in the four quadrants, although with some differences; a decrease in CLPs was found in 2/15 samples and a decrease in HSP/MPP populations was found in 8/15 samples. No aberrant antigen expression was observed in the CD38/all markers dotplots.

In Case 23, the 20-color panel showed a population in the LLQ that abnormally and strongly expressed CD38 and which showed a partial loss of CD200 (Figure 4). The three samples with pos MRD based on routine analysis (Cases 24–26) were also found to show abnormal HSPC populations in the 20-color panel (Figure 4). In all three samples, the ratio of myeloid to lymphoid HSPCs was increased; in Case 24, the blast population was centered in the LRQ and aberrantly expressed CD36; in Case 25, the blast population was found mainly in the RLQ, almost no normal HSPC populations were present in either upper quadrants, and CD200 was lost; and in Case 26, the blast population, although in the LLQ and LRQ, had also lost CD200 expression.

Three additional samples were studied from patients diagnosed as MDS, MDS/MPS, or CMML as well as seven cases with AML (see Appendix A). In all cases, the blast population clearly differed from the HSPC populations found in normal samples, and, in several cases, some markers were aberrantly expressed.

Samples from twelve patients with other hematologic diseases were used as controls (see Appendix A). In none of the twelve samples studied, aberrant marker expression was observed, and the distribution of HSPC populations in the CD371/CD45RA dotplot was found to be similar to normal BM samples.

The analysis of these different HSPC populations for the aberrant expression of certain markers, and a complete MRD analysis based on 2D dotplots, is repetitive and time-consuming, even for experienced cytometrists. It was, therefore, decided to use unsupervised automated methods for the analysis of the data files to investigate their potential use in diagnostics.

### 2.4. Dimensionality Reduction

Applying the t-SNE CUDA algorithm, files from the five normal bone marrow samples were first investigated using 1 × 10^6^ cells from each file for the analysis. The t-SNA algorithm clustered them into populations of various sizes, with myeloid cells being by far the most important population. HSPCs were split into two distinct populations, one corresponding to CD19pos HSPCs and the other to myeloid HSPCs (Figure 5A).

A detailed analysis of the twenty marker expression intensities for the different populations and the comparison with the populations in the 2D dotplots gated in Kaluza allowed twenty-four different cell populations to be distinguished (Appendix A). Of note, the smallest population, which formed a tight cluster and was distinguished in each sample, corresponded to CD117pos mast cells. These cells constituted between 0.02% and 0.09% of the five different samples. Gating on different populations, such as T cells, mature B cells, hematogones, and dendritic cells just to name a few, and comparisons with manually gated populations using 2D analysis resulted in highly correlated quantities of cell numbers (CC > 0.95% for all populations studied). Since HSPCs constituted only a minority of the total BM cells, and in order to highlight the different HSPC subpopulations more clearly, a second t-SNE CUDA analysis was performed on CD34pos HSPCs gated as shown previously in Figure 1A. The analysis of the obtained t-SNE plots for 4500 HSPCs from normal BMs again showed the separation of CD19pos HSPCs from myeloid HSPCs and distinguished between immature CD38neg HSCs/MPPs, more-differentiated CD38pos HSPCs, and the precursors differentiating into erythroid (CD36pos), monocytic (CD64pos), and myeloid (CD371pos) cell populations (Figure 5B). The analysis of the twenty different markers allowed the visualization of each marker intensity in the populations defined by the t-SNE plot (Appendix A).

The other available samples (Appendix A), analyzed previously with traditional 2D analysis, were then investigated in the same way using t-SNE CUDA. In all the samples, t-SNE plots showed CD19pos HSPCs exactly at the same position as for the normal BMs (Figure 6). Gating on this population allowed its quantification for all samples and showed values that highly correlated to the values determined using traditional 2D gating (CC > 0.99). Similarly, for myeloid HSPCs, defining a gate on the t-SNE presentations of normal BMs and applying it to all twelve control cases of other hematologic diseases showed that all the myeloid HSPCs were also found inside this gate (Figure 6A,D). Interestingly, this was also the case for 15/16 samples with a negative MRD status (Figure 6B). In Case 23, a population was found outside of the myeloid gate (Figure 6C, right plot). This was the sample in which routine cytometry negativity reported MRD, the molecular biology analysis reported positive MRD, and the traditional analysis of the 20-color panel tube also reported positive MRD.

Subsequently, the cases with positive MRD status (Figure 6C), the three cases of MDS, MDS/MPS, and CMML (Figure 7A), and the seven AML cases (Figure 7B) were investigated. In all of them, the malignant blast populations formed tight clusters which lied outside of the gate determined for the normal myeloid HSPCs. Moreover, in each case, the malignant blast population occupied a position on the t-SNE plot different from the position of the blast populations of all the other cases and did not overlap with any of them (Figure 7).

### 2.5. Unsupervised Clustering with FlowSOM

CD34pos HSPCs obtained from the same datasets and analyzed via t-SNE CUDA were subjected to analysis using the FlowSOM algorithm. Multiple iterations were carried out to optimize the clustering parameters, including the number of clusters and metaclusters (MC), ensuring the robustness of the obtained results. An illustrative outcome is presented for sixteen samples and ten metaclusters, delineating distinct HSPC subpopulations and facilitating inter-sample comparisons (Figure 8, Table 1). In the case of both normal bone marrow (BM) samples and MRDneg samples, comparable frequencies were observed across the ten metaclusters, with MC4 corresponding to lymphoid HSPCs and MC5 to myeloid ones. MC1, 2, 3, 7, 9, and 10 contained all different cell clusters of minor cell populations < 1%, and MC6 and MC8 contained CD123pos and CD371 strongly pos HSPCs, respectively. In the three MRD-positive cases as well as in the AML cases, either MC5 (Cases 2, 25, and 28), MC1 (Cases 24, 23, and 29), or MC2 (Cases 23 and 29) was clearly increased. In all these pathologic cases, aberrancies in MC frequencies were present and could be easily detected and quantified. Heatmaps allowed the complete phenotype of the MC corresponding to the blast cell populations to be evaluated. At the level of clusters, differences between normal and leukemic blast populations were also observed, with clusters increased in AML samples that contained only minor cell counts in the normal samples.

A detailed comparison between results from the t-SNE and FlowSOM algorithms showed that both allowed concordant conclusions and detected abnormal populations in the same samples with highly similar frequencies.

## 3. Discussion

Detailed analysis of the CD34pos HSPC population is of great importance for the understanding of this compartment in physiologic or pathologic conditions, such as, for example, during an infection or the malignant disease of the HSCP pool, such as leukemia or MDS. The aim of this work was four-fold: (1) to develop a new 20-color panel with markers allowing a detailed analysis of the HSPC compartment; (2) to present a refined strategy to detect abnormal HSPC subpopulations; (3) to show that results from a spectral cytometer are comparable to those obtained with a traditional cytometer; and (4) that algorithms for an unsupervised analysis compare favorably to results obtained from conventional manual gating.

This study, while limited in scope to a relatively small number of cases examined at a single center, serves as a proof of concept for the efficacy of analyzing a single 20-color panel on a spectral cytometer. Notably, the findings suggest that such an approach compares favorably to the analysis of multiple 8–13 color tubes on a traditional machine and facilitates in-depth scrutiny of rare cell populations. This strategy improves the “different from normal” approach in which cells in 2D dotplots are either found in positions where, in normal samples, cells are absent or absent from positions in which cells are normally present. It also improves the “leukemia-associated immunophenotype” (LAIP) approach, since the analysis of 20 markers in a single assay improves precision compared to an analysis with several tubes with different markers. Potentially, it could also improve sensitivity since HSPC subpopulations can be analyzed individually for aberrancies compared to a global analysis of the CD34pos HSPC population.

The advantages of spectral cytometers, as highlighted in previous research [29], include their cost-effectiveness and reduced sample material requirements compared to traditional cytometers. Additionally, the use of extended multicolor panels, as opposed to the currently prevalent 8–13 color panels, enhances precision by enabling the assessment of a greater number of markers in a single tube.

It was shown that the expression of markers on leukemic blasts can be analyzed in a straightforward way on CD38 dotplots, and that several of the most frequently expressed aberrant markers, such as CD7, CD56, CD19, and CD371, can be appreciated in a single assay. Furthermore, the flexibility of extending this panel to 24 colors opens the door to the incorporation of several additional markers, thereby expanding the depth and breadth of analysis. CD38 has been used previously as a marker to distinguish immature from more mature HSPCs, being typically absent in the most immature cells and then continually increasing in its expression when the HSPCs develop into more mature CMPs and GMPs [30]. It is important to note that this increase is gradual and, thus, does not permit the clearcut separation of cells into different subpopulations. Zeijlemaker’s group therefore used an indirect approach to separate CD38neg from CD38pos cells using either a fixed cut-off or mature erythrocytes as internal controls for CD38 negativity [31]. It is shown here that we can clearly gate the HSC/MPP population on the CD371/CD45RA dotplot as a population being double negative but expressing CD200 and CD133. Based on this strategy, the analysis of the different HSPC populations for the expression of aberrant markers becomes well defined. As shown for Case 23, MRD was detected, although the standard approach missed it. A larger study of samples with positive and negative MRD is needed to evaluate more clearly the sensitivity that can potentially be reached with this strategy. Only a comparison with results from molecular biology methods, such as dd RT-PCR, can show if it has a place in routine diagnostics.

As the two primitive HSPC markers, CD117 and CD133, were incorporated into the panel, analysis of AML samples with blast populations lacking CD34 expression also becomes feasible.

Including CD49f or CD90 as additional HSC markers might also help to improve the sensitivity and precision of the assay and more precisely distinguish HSCs from MPP populations. However, practically implementing this may pose challenges, particularly due to the extremely small numbers of these cell populations, especially in follow-up samples post-BM transplantation.

The application of the t-SNE algorithm to the sample data unveiled a series of interesting observations:Comprehensive Cell Population Analysis: When applied to entire bone BM samples, the algorithm effectively clustered all major cell populations, which were also identified by manual gating. Notably, it also identified rare populations, such as mast cells, constituting only 0.02% to 0.09% of all cells analyzed across five normal BM specimens.Concordant Results with Manual Gating: Gating and quantification of cell populations on t-SNE plots yielded highly comparable results to manual gating, with a correlation coefficient exceeding 0.95 in all cases.Consistent Population Positions: In parallel analyses of multiple samples, identical cell populations consistently occupied identical positions on the t-SNE plot. Aberrant populations, such as blast populations in AML, manifested in locations devoid of cells in normal BM samples, akin to the “empty space” regions in 2D gating.Distinct Positions for AML Blast Populations: Within AML cases, aberrant blast populations from different patients exhibited unique positions on the t-SNE plots, distinctly different from one another. Each blast population from the seven AML cases occupied a distinct position, contrary to traditional 2D analyses where blast cells often appeared devoid of aberrant markers, displayed a physiological phenotype, and were not distinguishable from normal HSPCs.

The significance of these findings lies in the t-SNE algorithm’s ability to position cell populations based on the integration of information from the twenty markers in the panel. Consequently, it can be inferred that subtle differences in the expressions of several markers within leukemic blast populations—undetectable at the level of individual markers through separate 2D dotplots—collectively contribute to their distinctive positions compared to normal HSPCs.

Moreover, these results suggest that by meticulously selecting a sufficiently high number of markers, malignant cell populations may be effectively differentiated from normal cells based on their distinct positions within a t-SNE plot. This approach holds promise for a wide range of analyses where the discrimination between malignant and normal populations is crucial, such as in various types of leukemias or lymphomas. These studies have started in my laboratory, paving the way for potentially new insights in the field. Our results on gated CD34pos HSPCs show that this approach is particularly promising for the detection of MRD in the follow-up of AML samples. In all the sixteen AML samples where MRD was negative, myeloid HSPCs were found in the same position on the t-SNE plot as in normal BMs and as in twelve control samples with other hematologic diseases. In the samples with positive MRD, the malignant blast population was clustered in a different position from normal and was specific for each patient.

Analysis with FlowSOM clustering led to results comparable to those obtained with t-SNE; abnormal populations were recognized with high fidelity in all pathologic samples with frequencies practically identical to those obtained using t-SNE or manual gating. In previous work, Lacombe et al. already described a similar approach several years ago based on the FlowSOM algorithm, but this was limited to nine color panels with partially manual clustering by cluster annotations and the need for a diagnostic sample, and it was impossible to track phenotypic shifts in the malignant cell populations [32].

In summary, this research has demonstrated the effectiveness of unsupervised and automated algorithms, notably t-SNE and FlowSOM, as promising alternatives to the laborious and time-consuming traditional 2D manual analyses of heterogeneous blood or bone marrow samples. This becomes particularly advantageous when dealing with a high number of markers. By employing panels that enable the concurrent analysis of 20 or more markers within a single tube, the accurate examination of even very rare cell populations, such as residual malignant blasts post-therapy, is greatly facilitated. Spectral cytometry serves as an optimal platform for implementing this multi-dimensional approach and establishing a robust automated and standardized analysis pipeline.

## 4. Material and Methods

### 4.1. Bone Marrow (BM) Aspirates

This study was reviewed and approved by the Institutional Review Board at University Hospital Geneva (Ethics Committee numbers 2020-00174 and 2020-00176). Five bone marrow (BM) samples from hip surgery patients without known hematological diseases were used to define lab-specific reference ranges for all considered FCM parameters. BM samples were also obtained from seven AML patients at diagnosis, nineteen follow-up samples after chemotherapy and/or BM transplantation (3 MRD pos; 16 MRD neg), three samples with malignant myeloid diseases (1 MDS, 1 CMML, and 1 MDS/MPS), and twelve samples from various other diseases (i.e., multiple myeloma, MGUS, etc.) (Appendix A). All BM aspirates were processed within 24 h of collection.

### 4.2. Routine Diagnostics for HSPC Analysis from BM Samples

For the routine diagnosis of AML as well as for controls post-chemotherapy/BM transplantation for MRD detection, the Routine Diagnostics Laboratory currently uses one to eight 10-color panels (Appendix A). The laboratory has developed gating strategies to analyze normal and aberrant CD34pos HSPCs and define MRD positivity with a sensitivity of 0.1%, as required by the ELN [16]. For normal HSPCs, myeloid CD33pos (approx. 70% of CD34pos HSPCs) and lymphoid CD19pos (approx. 30% of CD34pos HSPC) subpopulations are defined (Appendix A). Subsequently, normal and aberrant lineage marker expressions are analyzed on the myeloid subpopulation (Appendix A) using a combination of LAIP and “different from normal” approaches [33]. The panels are run on a Navios cytometer, with the acquisition of >1 million events, and analyzed using KALUZA software Version 2.1 (Beckman Coulter, Brea, CA, USA).

To determine MRD status, the results from flow cytometry are routinely compared to results from molecular biology analysis, and a consensus result is reported. In the sixteen cases where MRD was found to be negative using flow cytometry, one case gave a positive result using NGS for RUNX1 and SF3B1 mutations and, thus, was considered to be MRD positive (Case 23; Appendix A).

### 4.3. Panel Design and Sample Analysis

The fluorochrome combinations for the 20-color panel (CD3, CD7, CD14, CD15, CD16, CD19, CD33, CD34, CD36, CD38, CD45, CD45RA, CD56, CD64, CD117, CD123, CD133, CD200, CD371, and HLA-DR) were selected based on antigen density, fluorochrome intensity, expression profile, and reagent availability using Cytek’s Full Spectrum Viewer. The Complexity Index of our panel was 20.97. The composition (antibodies, clones, and fluorochromes) of the panel can be found in Appendix A. The cells were stained for 30 min at room temperature. To preserve the integrity of the fluorochromes, all staining procedures were performed in the dark, and to achieve optimal performance, all antibodies were titrated and used at saturating concentrations. A viability dye (Via Dye; Cytek) was added before analysis on a full-spectrum flow cytometer (Northern Lights, Cytek Biosciences, Fremont, CA, USA), and 1–1.5 × 10^6^ events were acquired to assess low-frequency populations, such as stem cells and progenitors, and for the reliable assessment of immunophenotypic aberrancies. To maximize the signal resolution of the measured fluorescent probes, “autofluorescence extraction” was enabled by default during sample acquisitions. Prior to analysis, the quality of the acquired and unmixed data was examined with SpectroFlo software version 1.0.2 (Cytek), and minor post-acquisition unmixing issues were manually corrected when the biology of the markers was known. Files were then exported in FCS format and analyzed using Kaluza software version 2.1 (Beckman Coulter). After gating out artifacts (Time/SSC), dead cells (viability dye/SSC), and doublets (SSC-A/SSC-H), files were uploaded onto the Cytobank^®^ platform (Beckman Coulter). Scales and compensations were corrected according to the Cytobank^®^ software (Version 10.3) instructions, and files were then analyzed with the t-SNE CUDA dimensionality reduction algorithm and the FlowSOM clustering algorithm. t-SNE CUDA is an implementation of the t-SNA algorithm, which uses GPU (graphics processing units) to reduce the computational time of the t-SNE algorithm [34].

## Figures and Tables

**Figure 1 ijms-25-02847-f001:**
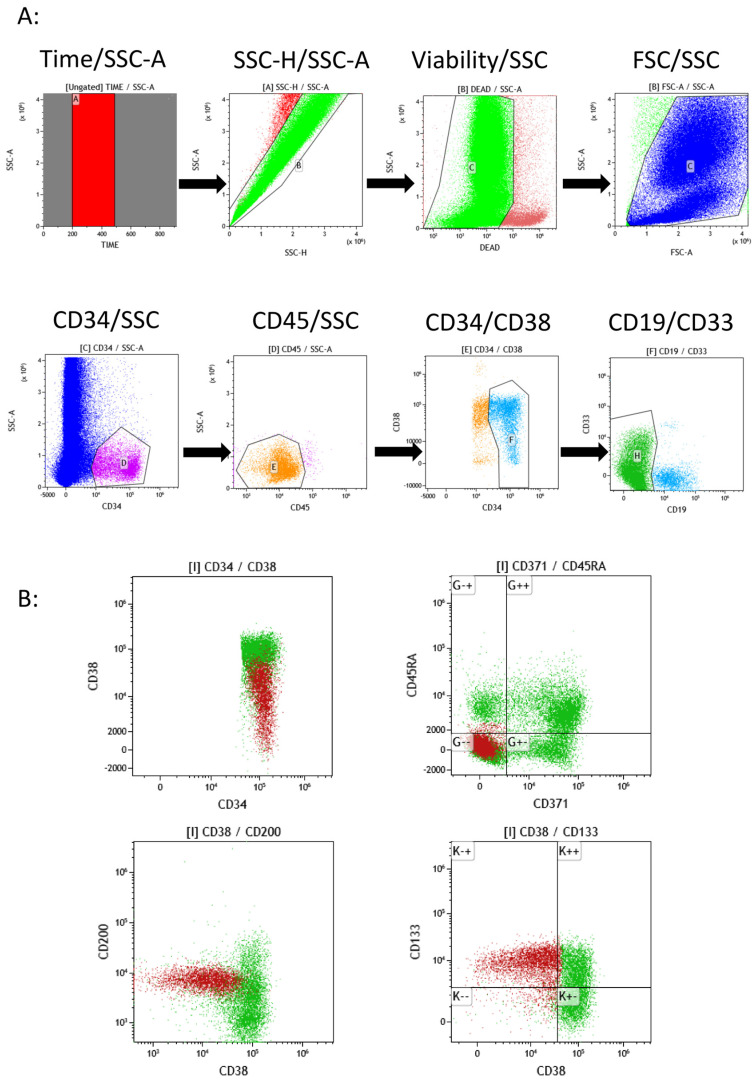
Gating strategy for the analysis of CD34pos HSPCs with the 20-color panel. (**A**) Gating of HSPCs using a sequential gating strategy of Time/SSC to eliminate acquisition artifacts, SSC-H/SSC-A to eliminate doublets, viability dye/SSC to eliminate dead and apoptotic cells, and CD45/SSC to eliminate non-lysed red blood cells and platelets. CD34pos cells were finally gated in the CD34/SSC dotplot. Backgating on CD45/SSC and CD34/CD38 helped eliminate unwanted non-HSPC cells in the gate. Analysis of the CD19/CD33 dotplot finally allowed lymphoid CD19pos (azure) to be distinguished from myeloid CD19 neg and CD33 neg/pos HSPCs (green). Capital letters in the different dotplots and in the titles define gates and gated populations, respectively. (**B**) Analysis of myeloid HSPC subpopulations. HSC/MPP subpopulations of HSPCs were found exclusively in the CD371neg/CD45RA neg quadrant of the CD371/CD45 2D dotplot (brown color). This population showed CD38dim/neg expression, in contrast to more mature subpopulations such as CMPs and GMPs/MEPs (upper left dotplot). In addition, this population showed a characteristic CD200 and CD133 positive expression, as shown in the two lower 2D dotplots. A typical example of the five studied normal BMs is shown.

**Figure 2 ijms-25-02847-f002:**
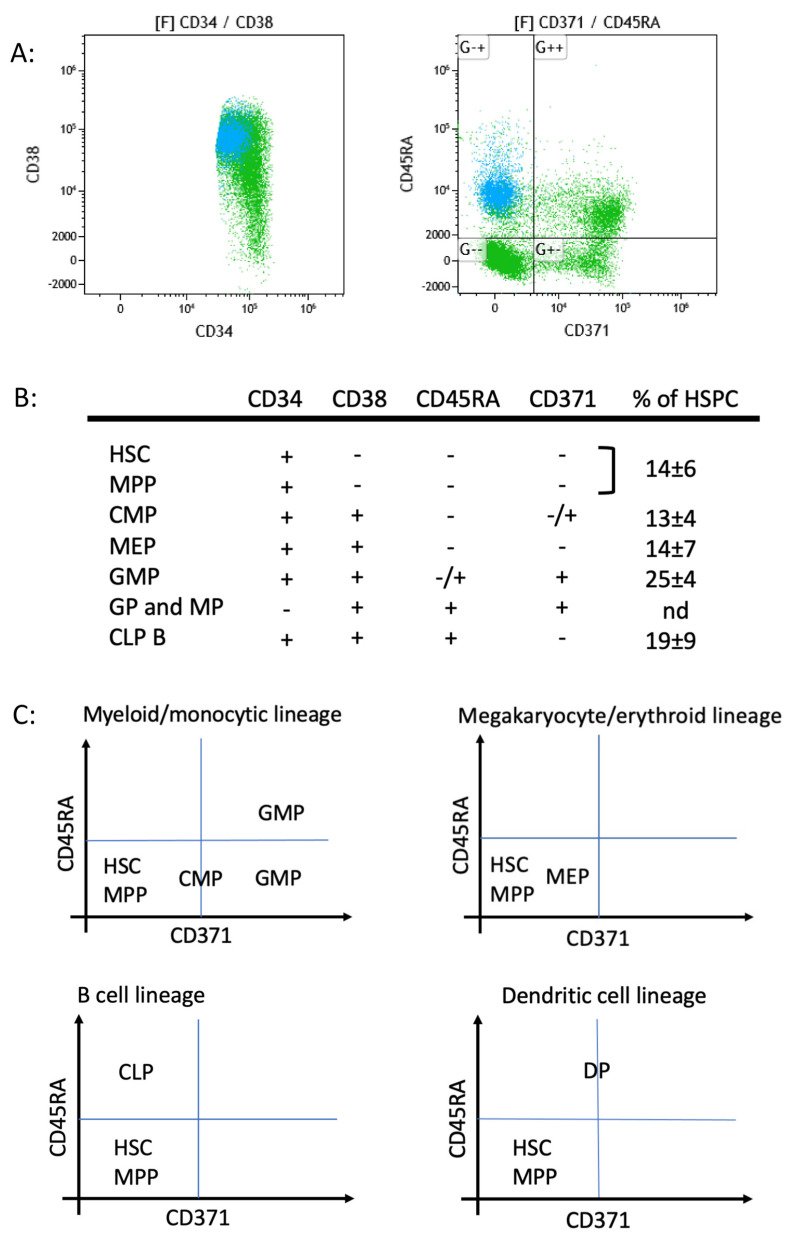
Dissection of HSPCs into different subpopulations. (**A**) Analysis of CD34pos HSPCs in CD371/CD45RA 2D dotplots allows clear separation into four different clusters. Additional analysis with CD34/CD38 and CD19/CD33 as well as other marker combinations allows CLPs (azure) from myeloid progenitors (green) to be distinguished. (**B**) Distinction of different HSPC subpopulations based on four markers: CD34, CD38, CD45RA, and CD371. Percentage (%) values (mean ± SD) are given, based on the analysis of five normal BM samples. (**C**) Different CD34pos HSPC subpopulations occupy different positions in the four quadrants of the CD371/CD45RA dotplot. HSC/MPP cells are always positioned in the lower left quadrant (LLQ).

**Figure 3 ijms-25-02847-f003:**
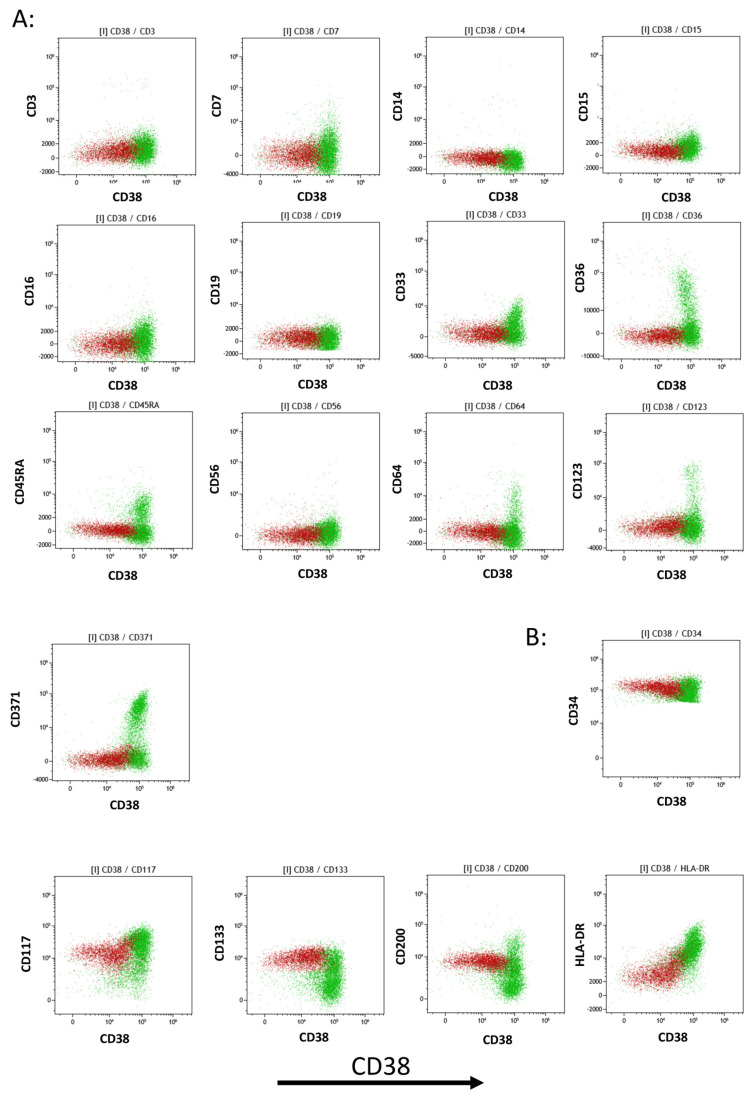
Analysis of co-expressed antigens on myeloid CD34pos HSPCs. (**A**) Analysis of myeloid HSPCs according to CD38 expression (*X*-axis) shows that the HSC/MPP population (brown color) did not express the following antigens: CD3, CD7, CD16, CD56, CD14, CD19, CD45RA, CD371, CD15, CD64, CD36, and CD123. Some of the more mature CD38pos HSPCs (green color) partially expressed CD64, CD36, CD123, and CD371, corresponding to cells starting to differentiate into monocytic, erythroid, dendritic, and myeloid cell populations, respectively. (**B**) Analysis of myeloid HSPCs according to CD38 expression (*X*-axis) shows that the HSC/MPP population (brown color) characteristically expressed CD200, CD133, CD117, and HLA-DR. A representative example from the five normal BMs studied is shown.

**Figure 4 ijms-25-02847-f004:**
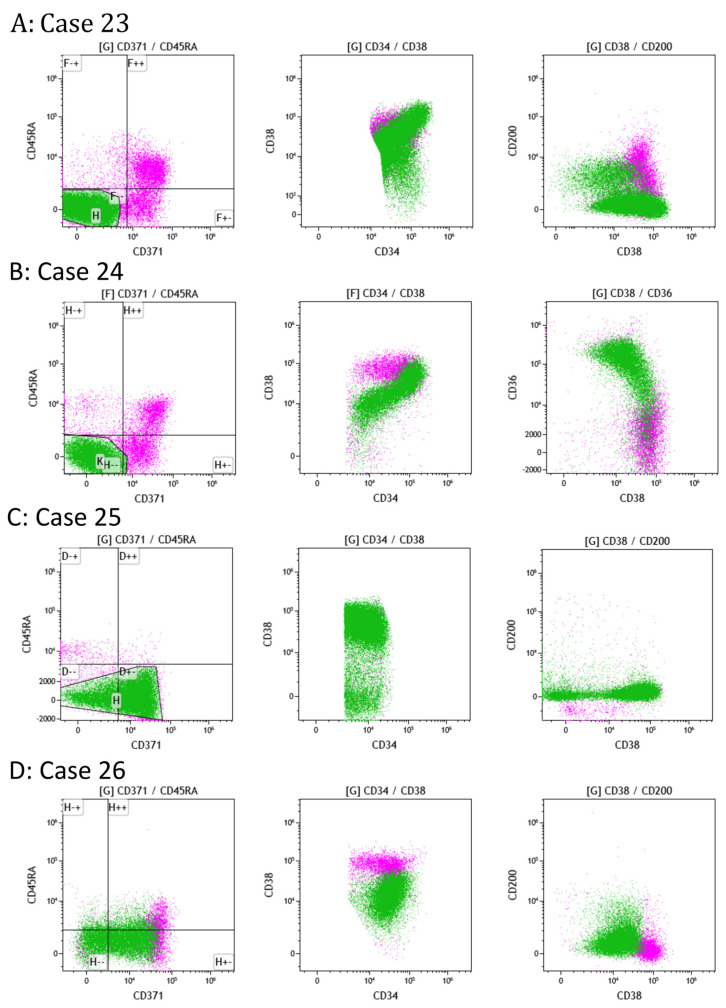
Analysis of CD34pos HSPCs from four different patient samples. The abnormal HSPC subpopulations are depicted in green (the remaining HSPCs are in pink), and three representative dotplots from each sample are shown. (**A**) Case 23: subpopulation in the LLQ of the CD371/CD45RA dotplot with normal HSCs/MPPs (CD200pos) and a larger, abnormal CD200 neg population. This population also abnormally expressed CD38. (**B**) Case 24: abnormal population in the LLQ of the CD371/CD45RA dotplot with abnormal CD36 expression and an abnormal position in the CD34/CD38 dotplot. (**C**) Case 25: abnormal population mainly in the LRQ with the loss of CD200 expression. Practically no cells were found in both upper quadrants. (**D**) Case 26: abnormal population in the CD371/CD45RA dotplot positioned in the LLQ but also the LRQ, with an abnormal position in the CD34/CD38 dotplot and the loss of CD200 expression.

**Figure 5 ijms-25-02847-f005:**
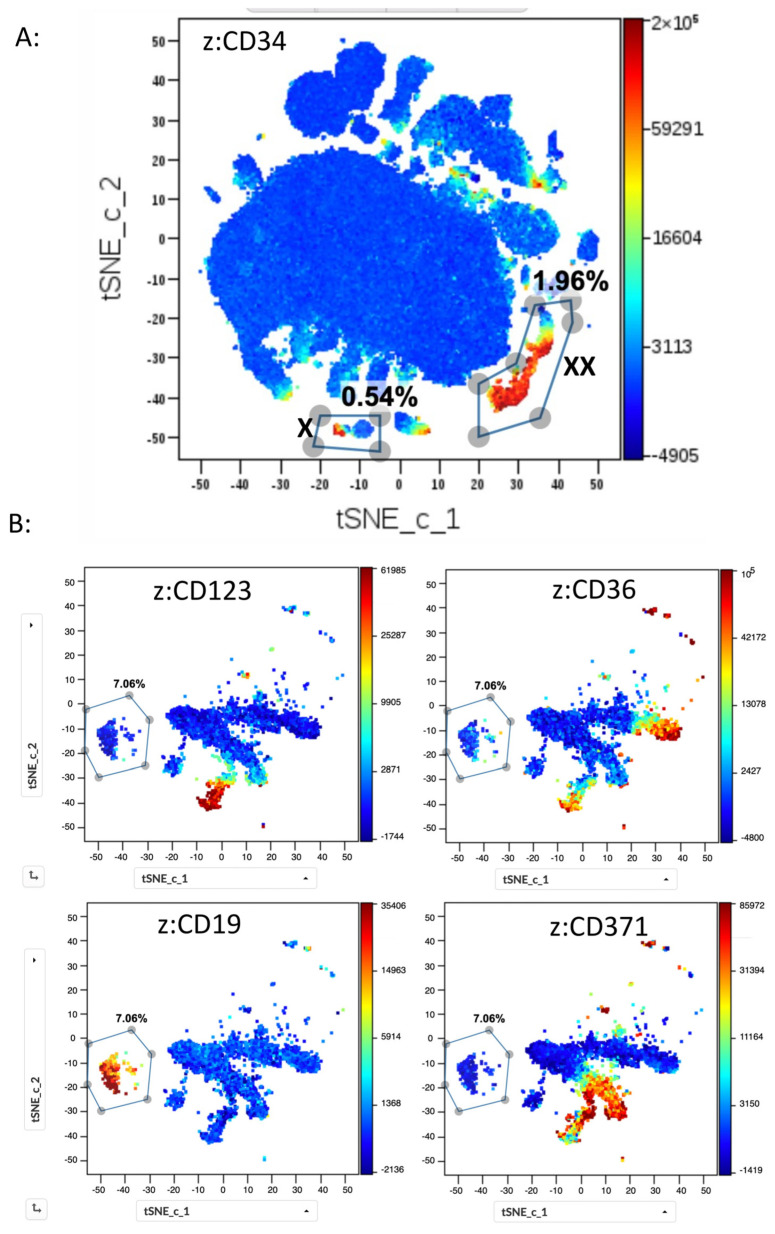
t-SNE CUDA analysis of CD34pos HSPCs from normal bone marrow samples. (**A**) Five normal BM samples stained with the 20-color panel were analyzed using t-SNE CUDA with 1 × 10^6^ cells from each sample used for the analysis (perplexity 30, iterations 1000; z channel: CD34). High similarity among the five samples is observed in the size and position of the different cell populations in the t-SNE plots. Shown is one typical example. Myeloid cells constitute by far the most important cluster, and CD19pos HSPCs (marked with **X**) occupy a position distinct from the myeloid HSPCs (marked with **XX**). (**B**) CD34 pos HSPC cells were gated according to the strategy described in Figure 1 and analyzed using t-SNE CUDA using 4500 cells from five normal BM samples for the analysis (perplexity 100, iterations 1000). Four representative plots from one patient are shown. The z-channel marker is marked for each plot. CD19pos precursors form a cluster distinct from the myeloid cluster. In the myeloid cluster, three different branches can be visualized, corresponding to cells differentiating into erythroid (CD36pos), dendritic (CD123pos), and myelomonocytic (CD371pos) cells.

**Figure 6 ijms-25-02847-f006:**
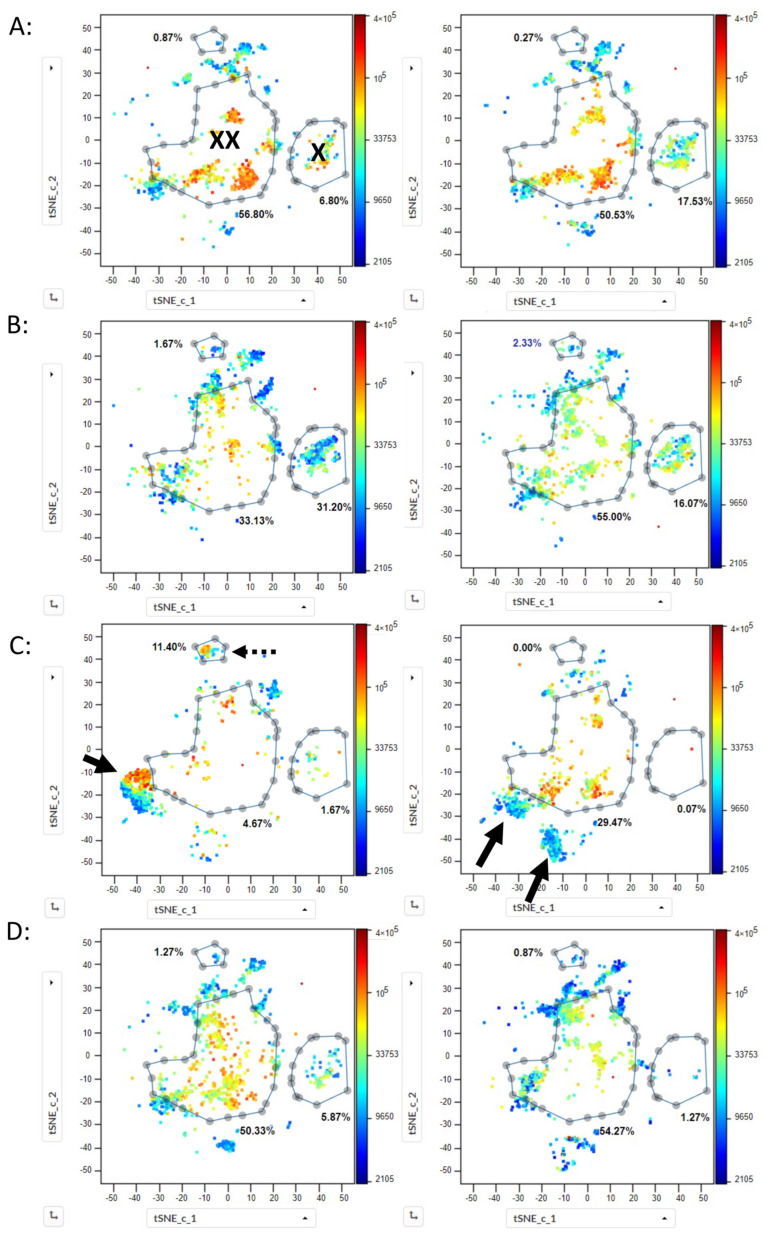
t-SNE analysis of CD34pos HSPC populations from different bone marrow samples. CD34pos HSPCs were gated according to Figure 1A and 1500 cells from each sample were used for the analysis using t-SNE CUDA (perplexity 100, iterations 1000, z channel: CD34). Arrows mark abnormal and malignant blast populations. (**A**) normal HSPCs from 2/5 normal bone marrow samples are depicted. Gates are drawn on CD19pos HSPCs (**X**) and on myeloid HSPCs (**XX**). Numbers correspond to the % of cells in the corresponding gates. A small gate in the upper part of the plot is also shown, which corresponds to CD56pos HSPCs. CD19pos and myeloid HSPC populations from the five normal samples lie exactly at the same position on the t-SNE plot. (**B**) HSPCs were analyzed on fifteen samples from AML patients with negative MRD after treatment. Two typical examples are shown (Case 20, left plot; and Case 22, right plot). Myeloid HSPCs are all located inside the myeloid HSPC gate drawn on the normal BM samples. (**C**) HSPCs were analyzed on three samples from AML patients with positive MRD after treatment. Two typical examples are shown. In Case 24 (left), one well-defined malignant blast population constituted 66% of all HSPCs; and in Case 23 (right), two abnormal populations lie outside of the gate of normal myeloid HSPCs (53%). Of note, in Case 24, a small, second abnormal population is found outside the myeloid gate but in the same spot as the CD56pos HSPCs in the other samples (dotted arrow; 11.4%). In all other samples, cells in this gate constituted < 3% of the HSPCs. (**D**) HSPCs of ten samples from patients with other hematologic diseases (Appendix A) were analyzed, with 2/10 samples shown—Case 30 (MGUS; left plot) and Case 40 (pancytopenia post liver transplant; right plot). Myeloid HSPCs in the two samples, as well as in the other eight samples, were always located inside the myeloid HSPC gate drawn on the normal BM samples.

**Figure 7 ijms-25-02847-f007:**
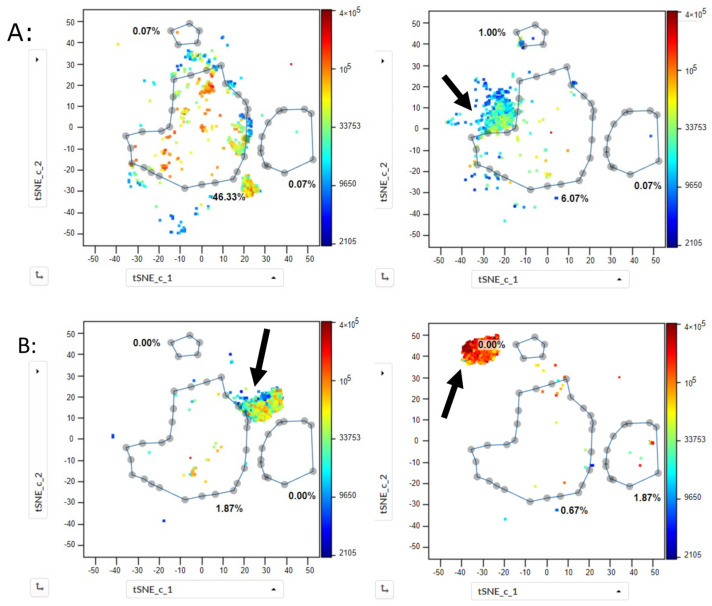
t-SNE analysis of CD34pos HSPC populations in bone marrow samples from AML, MDS, MDS/MPS, and CMML at diagnosis. HSPCs were gated in each sample according to Figure 1A and 1500 cells were used for the analysis using t-SNE CUDA (perplexity 100, iterations 1000, z channel: CD34). Same analysis as in Figure 6. Arrows mark abnormal and malignant blast populations. (**A**) CMML sample (Case 27, left) and MDS/MPL sample (Case 28, right). A well-defined blast population of myeloid HSPCs is visible in the CMML sample which lay outside of the gate for normal myeloid HSPCs (37%). In the MDS/MPS sample, an abnormal population (86%) is found outside of the gate for normal myeloid HSPCs. In the MDS sample (Case 29), 24% of the cells were found to be abnormal. (**B**) Shows 2/7 samples from patients with AML at diagnosis (Case 5, left; and Case 2, right). The malignant blast populations, 96% and 97%, respectively, are clearly defined and lie outside of the gate for normal myeloid HSPCs. Of note, both populations lie in different positions in the t-SNE plot.

**Figure 8 ijms-25-02847-f008:**
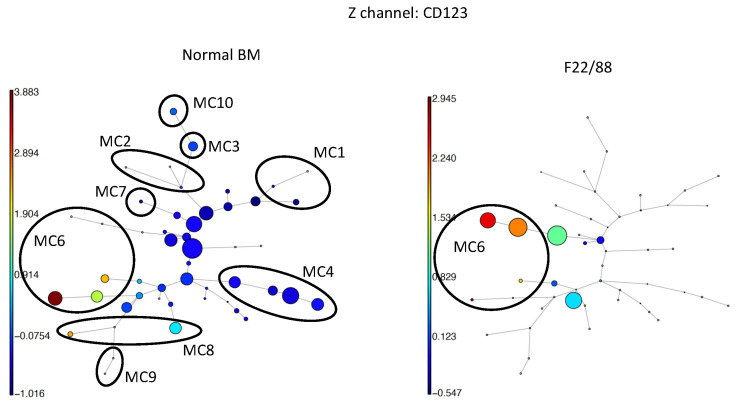
FlowSOM analysis. Shown are 2/16 examples of a FlowSOM analysis on CD34pos HSPCs with 1500 cells/sample; a normal BM sample (**left**) and Case 5 (AML; **right**). Forty-nine clusters and 10 metaclusters (MC) were chosen for classification. MC5 is made up of all the clusters not part of any of the other nine MCs. As z-channel, staining with CD123 was chosen. In the normal sample only 8% CD123pos cells were found in MC6 compared to 44% of the HSPCs in the AML sample (see Table 1). It can also be appreciated that in the AML sample, the two clusters that were increased (red and orange colors) contained practically no cells in the normal sample.

**Table 1 ijms-25-02847-t001:** FlowSOM analysis. CD34pos HSPCs from sixteen samples were analyzed using the FlowSOM algorithm with 1500 cells/sample; 49 clusters and 10 metaclusters (MC) were chosen for classification. Highlighted in yellow are the major clusters MC4 and MC5. Shown are the percentages for each of the ten MCs for each sample. Abnormally high values are marked in orange.

Metacluster		nBM	nBM	nBM	MRD neg	MRD neg	MRD neg	MRD neg	MRD pos	MRD pos	MRD pos	AML	AML	AML/MDS	MDS/MPS	CMML
					Case 20	Case 22	Case 14	Case 17	Case 24	Case 23	Case 25	Case 2	Case 5	Case 29	Case 28	Case 27
**1**	**minor pop (CD36pos)**	1	<1	<1	8	<1	4	1	38	19	<1	<1	<1	13	<1	<1
**2**	**minor pop (CD36pos)**	<1	2	<1	2	1	2	2	2	22	<1	<1	<1	11	<1	1
**3**	**minor pop (CD36pos)**	<1	10	<1	1	<1	5	<1	<1	<1	<1	<1	<1	<1	<1	<1
**4**	**normal lymphoid HSPC**	20	8	9	<1	24	3	42	2	<1	<1	1	<1	2	<1	6
**5**	**normal myeloid HSPC**	60	68	72	88	65	71	46	25	55	98	98	56	67	97	53
**6**	**DC precursors (CD123pos)**	8	8	9	<1	5	8	3	8	<1	<1	<1	44	4	<1	<1
**7**	**minor pop (CD7pos)**	<1	<1	<1	<1	<1	<1	<1	8	<1	<1	<1	<1	<1	<1	<1
**8**	**CD371pos**	5	3	6	<1	3	3	5	14	2	<1	<1	<1	<1	2	6
**9**	**minor pop (CD56pos)**	<1	<1	<1	<1	<1	<1	<1	<1	<1	<1	<1	<1	<1	<1	32
**10**	**minor pop (CD64pos)**	1	<1	<1	<1	<1	<1	<1	<1	1	<1	<1	<1	<1	<1	<1

## Data Availability

The raw data supporting the conclusions of this article will be made available by the author on request.

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
