# Peer review of "Phenotypic Analysis of Hematopoietic Stem and Progenitor Cell Populations in Acute Myeloid Leukemia Based on Spectral Flow Cytometry, a 20-Color Panel, and Unsupervised Learning Algorithms"

_ijms, 2024, doi:10.3390/ijms25052847_

Round 1

Reviewer 1 Report

Comments and Suggestions for Authors

The paper by Matthes et al is well written and conclusions well supported by the results. The technique describe and analysis of clinical samples are certainly interesting. It seems that such new strategy and large panel of antibodies for cytometry analysis should provide improved resolution of difficult to analyse patient samples. While the clinical samples used in this study are small in numbers, it remains significant. The figures are clear and adequate. Here are suggestions to improve the manuscript;

The author should make it clear if all figures presented are using the new 20-color panel and FSFC cytometer. Some time the “tube cytometer” are mention and its not clear then if results of classic FACS is shown or not, though I do not believe it is.  This can be done either in the fig legend or in the main body as see fit.

It is recommended to use neutral tense. As only one author is listed, it leads to the use of “I” and “my” which is unusual, and sometime a bit too much. I suggest that the author avoid using the term “my” by replacing it with “this” as it seems to work fine. As for replacing “I” in the text, it may be more challenging, but the author can keep some and substitute some with “shown in this work” or “in this work” or “in this study”.

Why did the author omitted the CD90 and CD49f antigens in their work given how useful they are in defining HSC and MPP? Maybe this could be stated in manuscript.

Page 3, Line 20; Title of subsection should be “Phenotypic analysis of normal HSPC subpopulations…” just to reinforce that this is not based on functional readouts,

Line 93; author should define abbreviation t-SNE

Page 4 fig 1: looks like the wrong dot-plot shown in A, top line, far right plot, seems like it should be CD45/SSC to remove red cells.

Page 5, Paragraph line 155-170; the author state that in their work that CD371 is not expressed by HSC/MPP. The author put a citation from A. Bakker et al., 2004 stating indeed that CD371/CLL-1 is not expressed by uncommitted CD34+CD38- HSPC. However, do the author have additional data supporting that the so-called LLQ in CD45RA/CD371 plots are really enriched in HSC/MPP? This claim could be substantiated with expression analysis within CD34+CD370-CD45RA- of other stem cell markers or lineage + events.

Page 6, Line 184; this sentence seems to be missing a word and is not making sense. “The markers in me panel…”

Page 7, Figure 3; it would be nice if the y (CDXX) and x (CD38) axes labels had bigger font cos its hard to read. The author could modify those in a different program.

Can the author provide a Table for the frequencies, SD or SEM, and median for the 5 distinct HSPC subsets (CLP, HSC/MPP, CMP, MEP and GMP) tracked in the 5 normal BM samples using the 20 color panel.

Author Response

Please find my answers in the attached Word file

Reviewer 2 Report

Comments and Suggestions for Authors

there is something wrong with figure 5: all lineages are shown with the same percentage and gate, I guess they all show the CD19 CLP committed part? CD38 is written instead of CD371.

do you think your panel would work in CD34 negative AML patients for follow-up?

we are never really informed where we use CD3-CD7-CD14-CD15-CD16-CD56-CD64-DR-117 in this 20-marker panel? after selecting the CD34 positive cells, they lose their meaning? Have they been used for following the differention paterns after HSPC state? will we read about them in another paper?

one wonders why you didn't include any ALL patients for comparison? would your panel also work in this setting? or would one need another panel to analyze B-CLP?

could a more narrowed-down panel be utilized to show true HSC size rather than the CD34 count in the ISHAGE? and could the conventional CD34 cut-off of 2x10^6/kg as the optimal limit be debated?

Author Response

Please find my answers in the attached Word file.
